# Loss of RNase J leads to multi-drug tolerance and accumulation of highly structured mRNA fragments in *Mycobacterium tuberculosis*

**Maria Carla Martini**[1☯¤], **Nathan D. Hicks**[2☯], **Junpei Xiao**[3], **Maria Natalia Alonso**[1], **Thibault Barbier**[2], **Jaimie Sixsmith**[2], **Sarah M. Fortune**[2]*, **Scarlet S. Shell**[1,3]*

**1** Department of Biology and Biotechnology, Worcester Polytechnic Institute, Worcester, Massachusetts, United States of America, **2** Department of Immunology and Infectious Diseases, Harvard T. H. Chan School of Public Health, Boston, Massachusetts, United States of America, **3** Program in Bioinformatics and Computational Biology, Worcester Polytechnic Institute, Worcester, Massachusetts, United States of America

☯ These authors contributed equally to this work.
¤ Current address: Institute of Biotechnology and Molecular Biology—CONICET, National University of La Plata, Buenos Aires, Argentina
* sfortune@hsph.harvard.edu (SMF); sshell@wpi.edu (SSS)

**Data Availability Statement:** Raw and processed RNAseq data are available in GEO, accession number GSE196357.

## Abstract

Despite the existence of well-characterized, canonical mutations that confer high-level drug resistance to *Mycobacterium tuberculosis* (Mtb), there is evidence that drug resistance mechanisms are more complex than simple acquisition of such mutations. Recent studies have shown that Mtb can acquire non-canonical resistance-associated mutations that confer survival advantages in the presence of certain drugs, likely acting as stepping-stones for acquisition of high-level resistance. *Rv2752c/rnj*, encoding RNase J, is disproportionately mutated in drug-resistant clinical Mtb isolates. Here we show that deletion of *rnj* confers increased tolerance to lethal concentrations of several drugs. RNAseq revealed that RNase J affects expression of a subset of genes enriched for PE/PPE genes and stable RNAs and is key for proper 23S rRNA maturation. Gene expression differences implicated two sRNAs and *ppe50-ppe51* as important contributors to the drug tolerance phenotype. In addition, we found that in the absence of RNase J, many short RNA fragments accumulate because they are degraded at slower rates. We show that the accumulated transcript fragments are targets of RNase J and are characterized by strong secondary structure and high G+C content, indicating that RNase J has a rate-limiting role in degradation of highly structured RNAs. Taken together, our results demonstrate that RNase J indirectly affects drug tolerance, as well as reveal the endogenous roles of RNase J in mycobacterial RNA metabolism.

## Author summary

*Mycobacterium tuberculosis* is the bacterium that causes tuberculosis (TB), which kills over a million people each year. Several antibiotics are effective against TB. However, *M. tuberculosis* frequently acquires mutations that cause antibiotic resistance, making treatment difficult and sometimes impossible. To develop better strategies to combat

**Funding:** This work was supported in part by NIH-NIAID award 5TP01AI143575-02 to SMF and SSS, by NIH-NIAID award U19 AI107774 to SMF, and by NSF-CAREER award 1652756 from the Directorate of Biological Sciences to SSS. The funders had no role in study design, data collection and analysis, decision to publish, or preparation of the manuscript.

**Competing interests:** The authors have declared that no competing interests exist.

antibiotic-resistant TB, we need to understand how resistance is acquired. Studies have revealed the presence of "stepping-stone mutations" that may cause bacteria to have low levels of antibiotic resistance, allowing some bacteria to survive treatment and acquire additional mutations that cause high levels of antibiotic resistance. Mutations in RNase J, a bacterial enzyme involved in the processing and degradation of RNA, were previously found to be associated with antibiotic resistance in *M. tuberculosis*. We hypothesized that these could be stepping-stone mutations and therefore investigated the relationship between RNase J and drug resistance. We found that deletion of RNase J causes more *M. tuberculosis* cells to survive antibiotic treatment. We determined that this increased survival is due to changes in gene expression that occur in the absence of RNase J. This work is important because it describes a new mechanism that bacteria can use to escape antibiotic treatment.

## Introduction

More than a century after the discovery that *M. tuberculosis* (Mtb) is the causative agent of tuberculosis (TB), this disease remains one of the major health challenges worldwide. In 2020, about 10 million people developed TB and 1.3 million died of the disease, positioning TB as a major cause of death worldwide [1]. Due to drug penetration issues and the presence of drug-tolerant Mtb populations in human lesions, anti-TB drug regimens must be administered over long durations [1–3]. Drug treatment itself may induce further drug tolerance [4], and the long treatment period provides opportunities for acquisition of mutations leading to antibiotic resistance. The emergence and spread of multidrug-resistant (MDR) TB are major concerns as it frequently leads to treatment failure and death. In 2020, 7.5% of the new TB cases were either rifampicin-resistant, MDR, or extensively drug resistant (XDR) [1] prompting the need to improve TB therapies and prevent the emergence of resistance.

Although the genes encoding the drug targets and activators in mycobacteria are well known, the mechanisms driving the development of high-level drug resistance are less well understood. In addition to high-level resistance, Mtb exhibits other forms of altered drug susceptibility that allow populations of bacteria to survive for extended periods of time in the presence of antibiotics and apparently serve as reservoirs for the eventual acquisition of high-level drug resistance-conferring mutations [5,6]. Initial studies on mycobacterial drug tolerance focused on rare persister cells that are drug-tolerant due to stochastic growth cessation ([7] and reviewed in [8,9]). Further work has demonstrated that altered drug susceptibility in Mtb can arise through multiple mechanisms, which differ in terms of frequency, duration, and magnitude of effect. For example, during infection Mtb responds to environmental and metabolic conditions by shifting to slow- or non-growing states in which it is less sensitive to many drugs [10–13]. Recently, several groups have taken a population genomics approach to identify clinically relevant stepping-stone mutations that facilitate the acquisition of high-level drug resistances. Mechanistic dissection of these mutations has revealed the importance of these different forms of altered drug susceptibility. For example, recent studies of Mtb have revealed unexpected forms of genetically-encoded low-level drug resistance such as mutations in *dnaA* and Rv0565c that cause low level isoniazid and ethionamide resistances, respectively [14,15]. Mutations in *prpR*, in contrast, were found to increase tolerance to multiple drugs without affecting resistance [16].

We and others have reported genome-wide association studies (GWAS) in cohorts of Mtb clinical isolates [16–19] and other bacterial pathogens [20–22] from around the globe. Many of

the Mtb studies have identified drug resistance associated mutations in Rv2752c, here called *rnj*, encoding the ribonuclease RNase J. This enzyme has both endonuclease and 5' to 3' exonuclease activity and is involved in the 5' end processing of ribosomal RNAs in *M. smegmatis* [23]. RNase J activity is essential for growth in several gram-positive bacteria that do not encode RNase E, in contrast to most gram-negative bacteria which encode RNase E and lack RNase J [24–26]. Unusually, mycobacteria encode both RNase E, which is essential and seems to play a rate-limiting step in degradation of most mRNAs [23,27–29], and RNase J, which is non-essential [23,27,29,30]. RNase J has been shown to participate in 23S rRNA processing [23] but we do not understand its impact on bacterial cell physiology and lack a model to mechanistically understand its relationship to drug responses. Here we investigate the role of RNase J in Mtb RNA metabolism and drug sensitivity. We show that *rnj* variants impact drug tolerance and mechanistically link the changes in drug susceptibility to altered gene expression and transcript degradation.

## Materials and methods

### Bacterial strains and growth conditions

*M. tuberculosis* H37Rv and its derivatives were grown in Middlebrook 7H9 broth supplemented with 10% OADC (0.5 g/L oleic acid, 50 g/L bovine serum albumin fraction V, 20 g/L dextrose, 8.5 g/L sodium chloride, and 40 mg/L catalase), 0.2% glycerol and 0.05% Tween 80. Liquid cultures were grown in 50 mL conical polypropylene tubes at 37°C with a shaker speed of 200 rpm except when indicated otherwise. For growth on solid media, Middlebrook 7H10 supplemented with 0.5% glycerol and OADC was used. For *M. tuberculosis* auxotrophic strain mc$^2$6230 ($\Delta panCD$, $\Delta RD1$, [31]) pantothenate was added to 7H9 or 7H10 to a final concentration of 24 μg/mL. When required for resistant bacteria selection or plasmid maintenance, the following concentrations of antibiotics were used: 25 μg/mL kanamycin (KAN), 50 μg/mL hygromycin (HYG) or 25 μg/mL zeocin (ZEO). Knock-out strains were constructed using the recombineering system described by Murphy and collaborators [32]. For genetic complementation, an L5-site integrating plasmid or Giles-site integrating plasmid was used, and for gene overexpression the episomal plasmid pMV762 was used [33]. A description of all strains used in this study is provided in S1 Table.

For growth of Mtb mc$^2$6230 in minimal media, log phase cultures grown in minimal media were sub-cultured to an $OD_{600nm} = 0.01$ in the same media. The minimal media had the following composition: 0.5 g/L asparagine, 1 g/L $KH_2PO_4$, 2.5 g/L $Na_2HPO_4$, 50 mg/L ferric ammonium citrate, 0.5 g/L $MgSO_4 \cdot 7H_2O$, 0.5 mg/L $CaCl_2$, and 0.1 mg/L $ZnSO_4$. Minimal media was supplemented with 0.05% Tween 80, 24 μg/mL pantothenate, and 0.2% glycerol.

### Antibiotic susceptibility testing

To determine the minimum inhibitory concentration (MIC) for Mtb mc$^2$6230 strains the agar proportion method was used [34]. Briefly, antibiotics were added to 7H10 plates to obtain the following concentrations: 1, 0.5, 0.25, 0.125, 0.06125, 0.0306, 0.01531 and 0.0077 μg/mL of rifampicin (RIF) or isoniazid (INH). Direct 5 μL aliquots and serial dilutions of 7H9 mid-log cultures of each strain were plated on antibiotic-containing and antibiotic-free plates. The MIC for each strain was determined as the drug concentration that reduced CFU by 90% compared to the control.

### Drug killing experiments

For Mtb mc$^2$6230 and H37Rv strains, log phase cultures grown in 7H9 were diluted to an initial $OD_{600nm} = 0.1$ in triplicate in absence of antibiotics. After 24 hours the following

antibiotics were added to the indicated final concentrations: 0.6 μg/mL of RIF, 2.4 μg/mL of INH, 2.5 μg/mL of clarithromycin (CLA), 1 μg/mL of ofloxacin (OFX), 2 μg/mL ethambutol (EMB), or 500 μg/mL erythromycin (ERY). Triplicate cultures of each strain were incubated in absence of drug as a control. Aliquots were periodically taken, and serial dilutions plated on 7H10 agar plates without drug. CFUs were counted after 20–35 days.

## Determination of the fraction of survival in INH at different growth phases

Log phase cultures of Mtb mc$^2$6230 WT or Δ*rnj* were sub-cultured to an $OD_{600nm}$ = 0.02. A total of 24 tubes were used per strain (6 replicates per timepoint). INH was added after 24, 48, 72, or 96 hours to a final concentration of 2.4 μg/mL. For each timepoint, CFUs were measured before adding INH (time 0 for each timepoint) and after 2 days of incubation with INH. The fraction of survival in INH was determined as the ratio between CFUs at day 2 over CFUs at time 0.

## RNA purification and quantitative PCR

For RNA purification from Mtb mc$^2$6230, frozen cultures stored at -80°C were thawed on ice and centrifuged at 4,000 rpm for 5 min at 4°C. For Mtb H37Rv, cultures were pelleted and processed immediately. The pellets were resuspended in 1 mL Trizol (Life Technologies) and placed in tubes containing Lysing Matrix B (MP Bio). Cells were lysed by bead-beating (2 cycles of 9 m/sec for 40 s, with 2 min on ice in between) in a FastPrep 5G instrument (MP Bio). 300 μL chloroform was added and samples were centrifuged for 15 min at 4,000 rpm at 4°C. The aqueous phase was collected, and RNA was purified using Direct-Zol RNA miniprep kit (Zymo) according to the manufacturer's instructions. For Mtb mc$^2$6230 samples, the optional on-column DNase treatment step was used. For H37Rv samples, in-tube DNase treatment was done using DNase Turbo (Ambion) followed by purification with a Zymo Clean & Concentrator kit.

For cDNA synthesis, 600 ng of RNA were mixed with 0.83 μL 100 mM Tris, pH 7.5, and 0.17 μL of random primers (3 mg/mL NEB) in a total volume of 5.25 μL. The mix was denatured at 70°C for 10 min and placed on ice for 5 min. For reverse transcription, the following reagents were added to achieve the specified amounts or concentrations in 10 μL reactions: 100 U ProtoScript II reverse transcriptase (NEB), 10 U RNase inhibitor (murine; NEB), 0.5 mM each deoxynucleoside triphosphate (dNTP), and 5 mM dithiothreitol (DTT). Samples were incubated for 10 min at 25°C and reverse transcription was performed overnight at 42°C. RNA was degraded by addition of 10 μL containing 250 mM EDTA and 0.5 N NaOH and heating at 65°C for 15 min, followed by addition of 12.5 μL of 1 M Tris-HCl, pH 7.5. cDNA was purified using the MinElute PCR purification kit (Qiagen) according to the manufacturer's instructions.

RNA abundance was determined by quantitative PCR (qPCR) using iTaq SYBR green (Bio-Rad) with 200 pg of cDNA and 0.25 μM each primer in 10 μL reaction mixtures, with 40 cycles of 15 s at 95°C and 1 min at 61°C (Applied Biosystems 7500). Primers used in this study are listed in S2 Table.

## Measurement of RNA half-life

For Mtb mc$^2$6230, 5 mL of biological triplicates of log phase cultures were treated with RIF (final concentration of 50 μg/mL) to inhibit transcription, as reported by [35]. After RIF addition, tubes were placed into liquid nitrogen after 0, 2, 5, 10, 20, 40, and 80 min. Cultures were frozen at -80°C until RNA purification. RNA was purified using Direct-zol kit (Zymo) as described above. Half-lives were calculated as previously described [36,37]. Timepoints 40 and 80 min were excluded from half-life calculations in cases where there were missing values due to low abundance or cases where they did not follow the initial exponential decay trend (S8 Fig).

## Construction and analysis of RNA expression and 5' end-directed libraries

For RNAseq libraries, 5 mL of log phase cultures of Mtb H37Rv WT, Δ*rnj*, and Δ*rnj::rnj*$_{OE}$ strains grown in 7H9 were placed in liquid nitrogen and stored at -80°C. RNA purification and construction of both RNA expression and 5' end-directed (non-5' pyrophosphohydrolase-converted) libraries were performed as previously reported [38,39]. For both types of libraries, Illumina HiSeq 2000 paired-end sequencing producing 50 nt reads was used. Sequencing was performed at the UMass Medical School Deep Sequencing Core Facility. Raw and processed data are available in GEO, accession number GSE196357.

## Bioinformatic tools and analyses

Reads from Mtb datasets were aligned to the NC_000962 reference genome using Burrows-Wheeler Aligner [40]. The FeatureCounts tool was used to assign mapped reads to genomic features, and DESeq2 was used to assess changes in gene expression in RNA expression libraries [41,42]. To differentiate fully upregulated genes from those having accumulation of reads in specific parts of the transcripts, we designed a pipeline (S1 Fig). Each gene was divided into non-overlapping 10-nucleotide (nt) windows and the mean read depth (coverage) in each window was calculated for Δ*rnj* and WT replicates, using the Bedtools Genomecov function with the -pc option to computationally fill in coverage between reads [43], and a custom Python script to compute the coverage for each 10 nt window. The log$_2$ ratio of coverage in Δ*rnj*/WT was computed for each window in each gene. Then, within each gene, the window with the median log$_2$ coverage was identified. DESeq2 also reports log$_2$ ratios for each gene, and the genome-wide mean of these log$_2$ ratios was close to zero as expected given the assumption that the majority of genes do not differ between strains. However, the genome-wide mean of the median 10 nt window log$_2$ ratios was -0.07. We therefore normalized the log$_2$ ratios of the median 10 nt windows for all genes by adding 0.07. Next, we determined the absolute difference between the DESeq2 log$_2$ fold change and the normalized median 10 nt window log$_2$ ratio for each gene. The standard deviation (SD) of these differences was then calculated. Among the genes reported as differentially expressed by DESeq2, we set a cutoff of two SDs, such that genes with differences of ≥2 SDs were classified as partially up or down-regulated, and genes with differences of <2 SD were classified as fully up or down-regulated. For genes that were partially up-regulated, the minimum free energy (MFE) and thermodynamic ensemble predictions of the overexpressed regions and adjacent upstream and downstream regions of the same length were determined using the RNAfold web server [44].

To specifically evaluate the characteristics of the 5' regions of RNA fragments that accumulated in the Δ*rnj* strains, the normalized read depths for 5' ends in the 5' end-directed non-pyrophosphohydrolase-treated libraries were compared. 5' ends were considered if they had a minimum read depth of five reads and the highest read depth in a 5-nt window (S6 Table). Those 5' ends with read depth ratios ≥10 in Δ*rnj*/WT were classified as enriched in the Δ*rnj* strain. A total of 381 5' ends were classified as enriched. For a comparison, we selected 1,000 5' ends that were equally represented in the WT and Δ*rnj* strains. The first 50 nt downstream of each 5' end were analyzed. The minimum free energy (MFE) of each 50-nt sequence was determined using the RNAfold web server [44], and the %GC was determined.

## Analysis of the sequence and structural contexts of mutations

Published crystal structures were used to visualize the structural contexts of RNase J point mutations using Mol* Viewer [45] on RCSB PDB. The structures used were *Deinococcus radiodurans* 4XWW and 4XWT [46], *Thermus thermophilus* 3T3O [47] and 3BK1 and 3BK2 [48], and *Streptomyces coelicolor* 5A0T [49]. To assess the sequence context similarity between Mtb

RNase J and the orthologs used for structural analysis, RNase J orthologs were aligned by Clustal Omega 1.2.4 [50].

## Results

### Mutations in RNase J are associated with drug resistance in clinical Mtb strains

We recently identified mutations in non-canonical drug resistance genes that are strongly associated with drug resistance in clinical Mtb strains isolated from China [16]. As mutations in these genes do not confer high-level drug resistance, we hypothesized that they may arise as intermediate steps in the acquisition of high-level drug resistance. One of the genes associated with drug resistance was Rv2752c, hereafter referred to as *rnj*, which encodes the bifunctional exo/endoribonuclease RNase J [23]. Genome-wide association studies (GWAS) revealed that mutations in *rnj* were associated with INH resistance in Mtb [16]. Our results were consistent with two previous GWAS studies that also identified *rnj* mutations in association with multi-drug resistance in Mtb [17,18].

In our data set, *rnj* was highly polymorphic in INH-resistant strains compared to drug sensitive strains and mutations were distributed throughout the CDS (Fig 1A, Table 1 and S3). In INH-resistant strains nonsynonymous mutations were the most prevalent ($\sim$80% of the total mutations), while frameshift and nonsense mutations were also found throughout the gene, suggesting selection for loss of function variants. We used crystal structures and mutational analysis data from three other bacterial RNase J orthologs to analyze the structural contexts of clinical Mtb *rnj* mutations (Figs 1B and S2) [46–49]. Several mutations caused non-conservative changes to residues involved in $Zn^{2+}$ coordination at the catalytic site (S3 Table) [46–49]. Several others introduced prolines predicted to disrupt α-helices or β-sheets in the C-terminal domain, which is required for dimerization and efficient cleavage in *Bacillus subtilis* [48]. Some mutations affected residues involved in RNA binding [46,47,49].

Bellerose and collaborators [51,52] recently found, using a TnSeq screening approach, that disruption of *rnj* increased survival of Mtb in mice treated with the clinical first line drug regimen. Taking this finding together with the observation of potentially deleterious mutations described above, we hypothesized that loss of RNase J function might confer a survival advantage to the bacterium in the presence of drug treatment, and that mutations in clinical strains arise as a part of an evolutionary pathway to the acquisition of high-level drug resistance.

### Loss of RNase J increases tolerance to several drugs

To investigate the link between RNase J and drug sensitivity in Mtb, we first constructed RNase J deletion ($\Delta rnj_{6230}$) and complemented strains ($\Delta rnj_{6230}::rnj$) in the Mtb mc$^2$6230 background ($\Delta panCD$, $\Delta$RD1) [31]. To determine the effects of RNase J on drug sensitivity, we assessed both MICs and bactericidal activities of clinically relevant drugs. The MICs for RIF and INH were the same for the $\Delta rnj_{6230}$ strain and its WT parent (0.125 μg/mL for both drugs), indicating that loss of RNase J does not confer resistance to either drug. However, the $\Delta rnj_{6230}$ strain displayed increased survival in the presence of lethal concentrations of RIF and INH compared to the WT and the complemented strains (Fig 2A), suggesting that the absence of RNase J leads to increased drug tolerance. We observed a similar behavior when exposing the strains to lethal concentrations of EMB, OFX, CLA, and ERY (Fig 2A).

We sought to verify these results in virulent H37Rv, constructing RNase J deletion ($\Delta rnj_{H37Rv}$) and complemented strains ($\Delta rnj_{H37Rv}::rnj$). We tested killing by RIF, INH, and OFX, and found that in each case, the $\Delta rnj_{H37Rv}$ strain had a survival advantage compared to

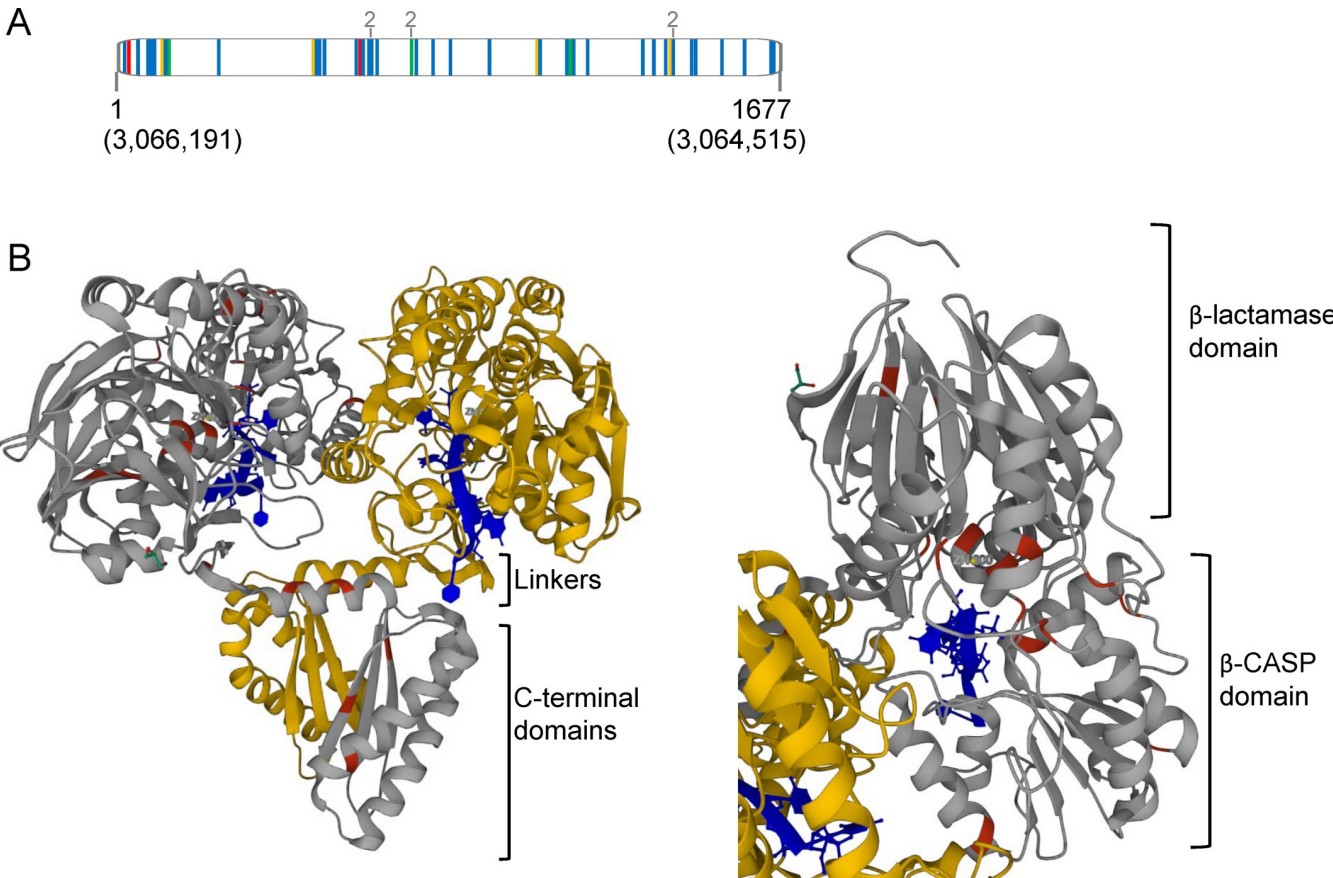

**Fig 1. RNase J is mutated in many clinical Mtb strains. A.** Vertical lines indicate mutations identified in clinical isolates in Hicks et al., 2018. Frameshift (red), nonsynonymous (blue), synonymous (orange), and nonsense (green) mutations are highlighted. Positions in the Mtb H37Rv genome are indicated in parentheses. Numbers in grey indicate mutations that evolved twice independently. **B.** Point mutations in INH-resistant clinical Mtb strains modeled on a *Thermus thermophilus* crystal structure (PDB 3T3O, Dorleans et al., 2011). Gold and gray indicate the two monomers, with Mtb mutated residues indicated in red on the gray monomer. A 4-mer RNA is indicated in blue. A catalytic zinc is indicated by "Zn." The second catalytic zinc is not present in the structure due to an active-site mutation needed to capture a stable RNA co-complex (Dorleans et al., 2011). Structures were annotated on RCSB PDB by Mol* Viewer (Sehnal et al., 2021). The view on the left is oriented to best visualize mutations in the C-terminal domain while the view on the right is oriented to best visualize the mutations in the catalytic region.

the WT and complemented strains (S3 Fig). Furthermore, complementation of $\Delta rnj_{\text{H37Rv}}$ with $rnj^{\text{H86A}}$, which is predicted to be catalytically dead [48], did not restore the WT sensitivity phenotypes for RIF, INH, or OFX. While we cannot exclude the possibility that Mtb RNase J with the catalytic site mutation is unstable, these data are consistent with the idea that the greater survival of the $\Delta rnj_{\text{H37Rv}}$ strain is due to loss of RNase J catalytic activity (S3 Fig).

To further characterize the survival advantage conferred by loss of *rnj*, we exposed cultures to a combination of drugs (INH and OFX) at lethal concentrations to prevent outgrowth of resistant mutants. We observed that both strains reached a plateau at a similar number of CFUs (Fig 2B), suggesting that loss of RNase J does not affect persistence but rather increases drug tolerance by reducing the killing rate. However, we cannot exclude the possibility that RNase J affects persister levels, since detection and quantification of persister cells are affected by the choice of assay and culture conditions (many works including [7,53–56]).

We noted that $\Delta rnj_{6230}$ colonies were smaller than those of the WT and complemented strains. We therefore measured the growth characteristics of the three strains (Fig 3A). The growth rate of $\Delta rnj_{6230}$ in mid-log phase was statistically indistinguishable from the growth

**Table 1. Summary of *rnj* mutations identified in clinical strains of *M. tuberculosis* and their predicted consequences.**

| Mutation type/region | Number of independent acquisitions | Predicted consequence | References |
|---|---|---|---|
| INH resistant strains (32 independent mutations) | | | |
| Frameshift | 2 | Loss of function | N/A |
| Nonsense | 4 | Loss of function | N/A |
| Catalytic site (non-conservative mutation) | 4 | Loss or reduction of function | [46–49] |
| RNA contacting region (non-conservative mutation) | 2 | Loss or reduction of function | [46–49] |
| Introduction of prolines to alpha helices or beta sheets | 5 | Disruption of alpha helix or beta sheet | [46–48] |
| Charge swap | 1 | Unknown | N/A |
| Change of hydrophobic residue to polar/charged, or change of polar/charged residue to hydrophobic. | 12 | Unknown | N/A |
| Synonymous | 2 | None | N/A |
| INH sensitive strains (9 independent mutations) | | | |
| Change of hydrophobic residue to polar/charged, or change of polar/charged residue to hydrophobic. | 3 | Unknown | N/A |
| Change of charged residue to polar or vice versa. | 4 | Unknown | N/A |
| Change of hydrophobic residue to hydrophobic residue. | 1 | Unknown | N/A |
| Synonymous | 1 | None | N/A |
| Strains with mixed INH resistance and sensitivity (4 independent mutations) | | | |
| Catalytic site (non-conservative mutation) | 1 | Loss or reduction of function | [46–49] |
| RNA contacting region (conservative mutation) | 1 | Unknown | [46–49] |
| Change of hydrophobic residue to hydrophobic residue. | 1 | Unknown | N/A |
| Synonymous | 1 | None | N/A |

rates of the WT and complemented strains. However, $\Delta rnj_{6230}$ had a slightly longer lag phase than WT cells. The lag phase delay was condition-dependent, as we did not observe it when growth curves were started at a higher OD (S4A Fig), or when the strains were grown in minimal media (S4B Fig). To test the possibility that the better survival of $\Delta rnj$ was due to the subtle growth defect observed at low ODs, we sub-cultured $\Delta rnj_{6230}$ and WT cultures to a low OD, added INH at a lethal concentration after one, two, three, or four days, and compared the fraction of survival after two days of incubation with drug (Fig 3B). We observed that the higher fraction of survival for $\Delta rnj_{6230}$ was consistent regardless of the growth phase at which drug was added (Fig 3C; growth phases spanned lag phase through late log phase). In addition, we performed RIF and INH time-killing curves in minimal media, where $\Delta rnj_{6230}$ does not show a longer lag phase, and found that $\Delta rnj_{6230}$ had better survival than WT in this condition as well (S4C Fig). While we cannot exclude the possibility that a longer lag phase contributes to the tolerance of $\Delta rnj_{6230}$ to some drugs, these data suggest that unlike what was previously reported in *E. coli* in response to intermittent treatment with ampicillin [57], lag phase differences do not explain the increased tolerance of $\Delta rnj_{6230}$ to continuous exposure to lethal concentrations of INH or RIF.

## RNase J affects rRNA processing and expression of a subset of genes and mRNA fragments

To investigate how RNase J mediates drug tolerance in Mtb, we assessed its role in RNA metabolism by performing RNAseq expression profiling as well as RNA 5' end mapping. These studies were done with $WT_{H37Rv}$ and $\Delta rnj_{H37Rv}$ strains transformed with the empty vector pJEB402 as well as the $\Delta rnj_{H37Rv}$ strain complemented with *rnj* under a strong constitutive

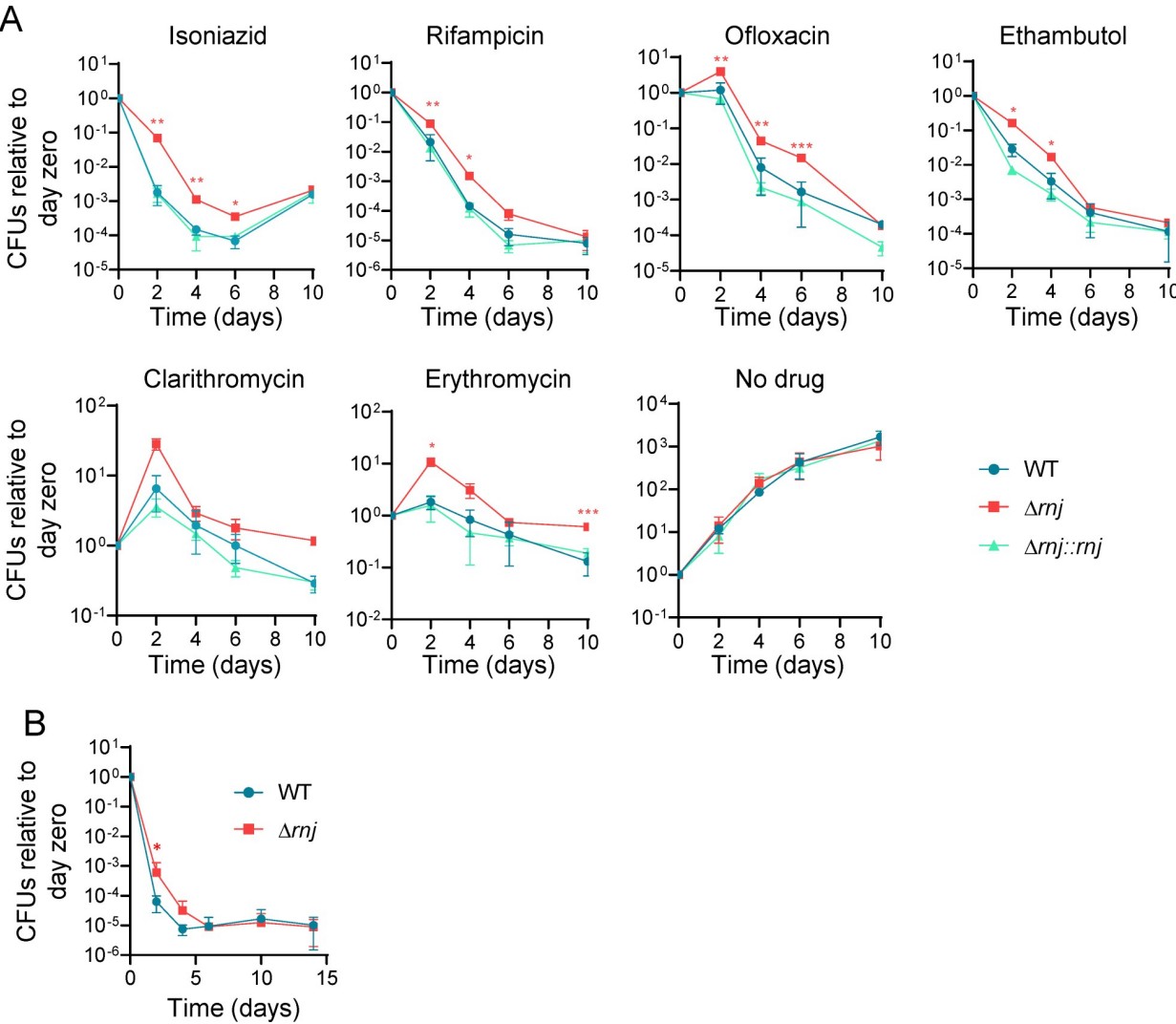

**Fig 2. Loss of RNase J increases tolerance to several drugs. A.** Time-kill curves in presence of the indicated drugs. The concentrations used were: 0.6 μg/mL RIF, 2.4 μg/mL INH, 2.5 μg/mL CLA, 1 μg/mL OFX, 2 μg/mL EMB, or 500 μg/mL ERY. **B.** Time-kill curves in presence of lethal concentrations of both INH (2.4 μg/mL) and OFX (2 μg/mL). Both experiments were performed using Mtb mc$^2$6230 strains. $^*p<0.05$, $^{**}p<0.01$, $^{***}p<0.001$ two-way ANOVA for comparisons of Δ*rnj* to WT. FDR 0.05 (Benjamini and Hochberg). Curves are representative of at least two independent experiments.

promoter, Δ*rnj*$_{H37Rv}$::*rnj*$_{OE}$ [58]. As a quality control, we evaluated the 23S rRNA as previous work has shown that RNase J is necessary for 23S rRNA maturation in *M. smegmatis* [23]. Consistent with these data, we found that the 23S rRNA transcript was 15 nt longer in Δ*rnj*$_{H37Rv}$ compared to WT$_{H37Rv}$ (S5 Fig), indicating that RNase J also plays a role in 23S rRNA processing in Mtb.

Standard analysis of the RNAseq expression data using DESeq2 indicated that 57 and 16 genes had increased or decreased transcript abundance, respectively, in Δ*rnj*$_{H37Rv}$ compared to WT$_{H37Rv}$ (fold change $\geq1.5$, adj *p* value $\leq0.01$) (Fig 4A, S4 Table). Comparison of changes in transcript abundance in the Δ*rnj*$_{H37Rv}$ and Δ*rnj*$_{H37Rv}$::*rnj*$_{OE}$ strains showed a significant negative correlation (S6 Fig), reflecting the opposite effects of *rnj* overexpression and deletion. However, visual inspection of RNAseq expression library coverage revealed that several differentially abundant transcripts did not have increased abundance across the entire gene in

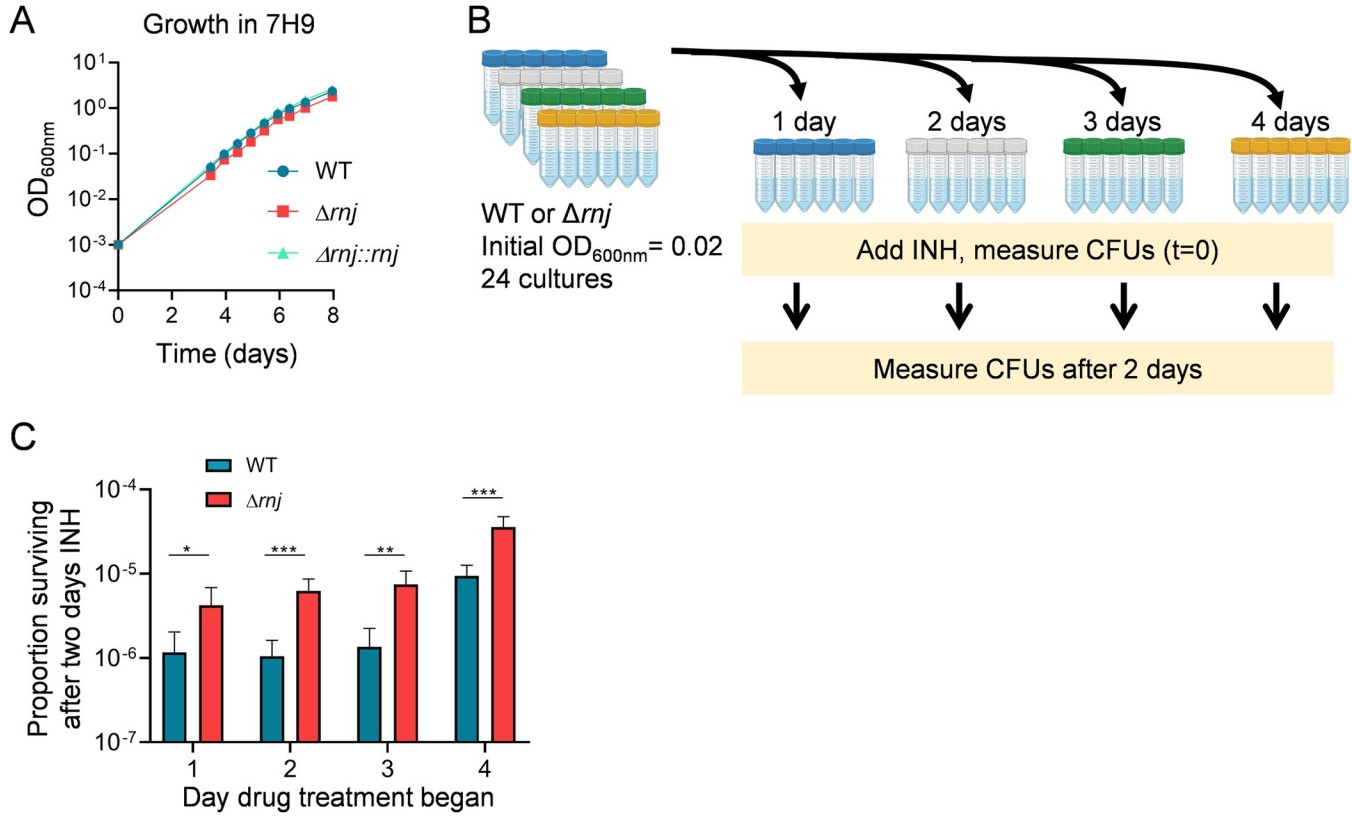

**Fig 3. Drug tolerance in Δ*rnj* Mtb is not due to its lag phase growth defect. A.** Growth kinetics of Mtb mc$^2$6230 WT, Δ*rnj*, and Δ*rnj::rnj* in 7H9 media. Slopes were statistically equivalent for all three strains in mid-log phase (days 3–6, linear regression). ODs were significantly different for Δ*rnj* vs WT for all time-points from day 3 onward (t-tests with FDR 1% correction) **B.** Schematic of experiment to determine the effect of growth phase on drug survival. Created with Bio.Render.com. **C.** Data from the experiment shown in B. Bars indicate the proportion of cells that survived after two days of incubation with INH (2.4 μg/mL) starting at the indicated days. This proportion is the value in the lower yellow bar in B divided by the value in the upper yellow bar in B. $^*p<0.05$, $^{**}p<0.01$, $^{***}p<0.001$, t-tests with Benjamini and Hochberg FDR 0.05.

Δ*rnj*$_{H37Rv}$, but rather displayed increased read coverage for only short segments of the genes in question (Fig 4B). We hypothesized that these short RNA fragments might have arisen from incomplete degradation of mRNAs in the absence of RNase J. Thus, standard DESeq2 analysis is not sufficient to discriminate between genes for which the abundance of the whole transcript changes and genes marked by the accumulation of RNA fragments. To address this issue, we developed a bioinformatics pipeline (see Materials and Methods and S1 Fig) to identify genes for which the abundance of the whole transcript was altered. Of the 57 genes that were reported as significantly overexpressed in Δ*rnj*$_{H37Rv}$ by DESeq2, 31 reflected increases in the whole transcript (Fig 4A, red dots and S4 Table) while the remaining genes had increased abundance of only subsections of their transcripts (Fig 4A, light pink dots, and S4 Table). All the genes with reduced abundance in Δ*rnj*$_{H37Rv}$ except for one (Fig 4A, light blue dot) showed changes in the abundance of the entire transcript (Fig 4A, blue dots, and S4 Table).

## RNase J has a specialized role in degrading mRNA fragments with strong secondary structure and high G+C content

We hypothesized that the transcript fragments that accumulated in the Δ*rnj*$_{H37Rv}$ strain could be RNase J targets that were inefficiently degraded in the mutant strain. To test this, we chose four genes with partial-transcript abundance increases in the Δ*rnj*$_{H37Rv}$ strain and measured

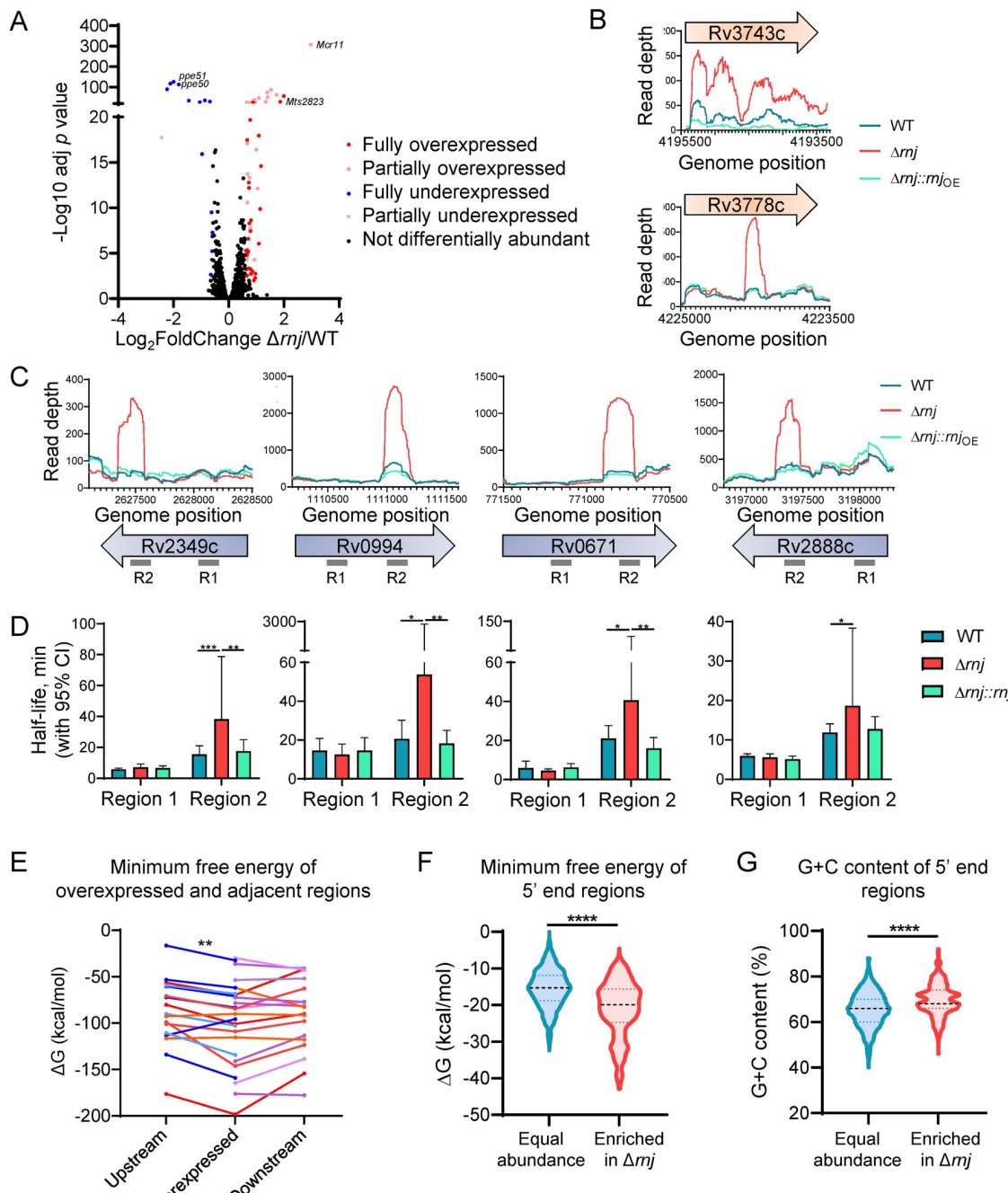

**Fig 4. RNase J affects expression of genes and causes highly structured mRNA fragments to accumulate in Mtb. A.** Volcano plot showing the genes affected by RNase J in Mtb H37Rv strains. Partially and fully over/under expressed genes are distinguished with different colors. **B.** Schematics of the read depth of two genes presenting full overexpression (upper panel) or partial overexpression (lower panel) in Δ*rnj*. **C.** Read depth of expression libraries in Mtb H37Rv strains for four genes that displayed partial overexpression in Δ*rnj*. Grey lines below the arrows denote the sequences targeted by qPCR for regions of the genes displaying accumulation of short fragments in Δ*rnj* (R2) and regions with similar read coverage in all strains (R1). For all H37Rv experiments, the WT and Δ*rnj* strains contained the empty vector pJEB402. **D.** Determination of half-life for the gene regions shown in C using Mtb mc²6230 strains. $^*p<0.05$, $^{**}p<0.01$, $^{***}p<0.001$. **E.** The minimum free energy of folding was predicted for overexpressed regions in A and for regions of equal lengths immediately upstream and downstream of each overexpressed region. Upstream and downstream MFEs were only calculated when the region fell within the coding sequence of the gene. Blue indicates overexpressed regions for which only the upstream adjacent region was available, purple indicates regions for which only the downstream adjacent region was available, and red indicates regions for which both upstream and downstream regions were available. $^{**}p<0.01$, Wilcoxon matched-pairs signed rank test. **F** and **G.** 5' end-mapping libraries were used to identify transcripts overexpressed in Δ*rnj*.

The 50 nt downstream of both overexpressed and unchanged 5' ends were analyzed to predict the minimum free energy of secondary structure formation and determine G+C content. ****$p<0.0001$, Mann Whitney test.

the half-lives of both the overrepresented regions of each transcript and regions for which read coverage was similar in both strains (Fig 4C). The overrepresented regions had longer half-lives in the absence of *rnj*, while half-lives of equally abundant regions did not differ between the strains (Fig 4D).

To understand why RNase J has an apparently rate-limiting role in degradation of certain mRNA fragments, we calculated the predicted minimum free energy of folding ($\Delta G$) for each overrepresented region (S5 Table) and for adjacent upstream and downstream regions of equal size, when those regions were present within the same coding sequence. The $\Delta G$s of the overrepresented regions were in most cases lower than the $\Delta G$s of the upstream adjacent regions (Fig 4E, $p = 0.0021$), indicating that formation of secondary structure was more energetically favorable for the overrepresented regions. There was an apparent trend toward the $\Delta G$s of the overrepresented regions also being lower than those of the downstream adjacent regions, but this was not statistically significant. We repeated the analysis using the free energies of thermodynamic ensemble predictions [44] and the results were nearly identical ($p = 0.0017$ for accumulated regions vs upstream regions). Inspection of the minimum free energy structures did not reveal any trends regarding the types and locations of predicted secondary structure elements. The library construction method did not permit identification of the exact 5' and 3' ends of each overrepresented region, which precluded analysis of the base-pairing probabilities of those ends.

Since the overrepresented regions appeared to be more structured than adjacent upstream regions and RNase J is known to have 5' to 3' exonuclease activity, we hypothesized that RNase J may contribute to degradation of transcripts that have high levels of secondary structure near their 5' ends. We therefore used a separate 5'-end-directed RNAseq dataset to analyze the properties of 5'-end-adjacent sequences in WT$_{H37Rv}$ and $\Delta rnj_{H37Rv}$ transcriptomes. We assessed the G+C content and predicted secondary structure of the 50 nt sequences adjacent to RNA 5' ends that had increased abundance in the $\Delta rnj_{H37Rv}$ strain or were equally abundant in the $\Delta rnj_{H37Rv}$ and WT$_{H37Rv}$ strains (S6 Table). We found that the 5'-end-adjacent sequences with increased abundance in $\Delta rnj_{H37Rv}$ had significantly more negative predicted minimum free energies of folding ($\Delta G$) than those that were equally represented in $\Delta rnj_{H37Rv}$ and WT$_{H37Rv}$ (Fig 4F). In addition, the median G+C content of the sequences overrepresented in $\Delta rnj_{H37Rv}$ was significantly higher than that of the equally represented sequences (Fig 4G). These results are consistent with the idea that RNase J targets RNAs with relatively strong secondary structure, particularly near their 5' ends.

Having found that loss of RNase J increased stability of short RNA fragments, we wondered if the increased abundance of the fully overexpressed genes in $\Delta rnj_{H37Rv}$ was a direct consequence of slower degradation rates or increased transcription. We measured the half-lives of six fully overexpressed genes and found that transcript stability was not altered (S7A and S8 Figs). However, these genes did not have high G+C contents, and technical difficulties prevented us from obtaining high-confidence half-lives for those fully overexpressed genes that were GC-rich (S7B Fig). There may therefore be genes that were fully overexpressed due to reduced degradation rates, and this should be addressed in future work. We also considered the possibility that the absence of RNase J could lead to accumulation of antisense transcripts that could affect mRNA degradation or translation. However, there were no changes in median antisense coverage for genes differentially expressed in $\Delta rnj_{H37Rv}$ or for all expressed

genes in this strain. Together, these data suggest that RNase J affects the transcript abundance of some genes through altered transcription rather than by altering their mRNA stability.

### Genes differentially expressed in the absence of RNase J are enriched for sRNAs, PE/PPE family genes, SigM targets, and genes with roles in hypoxia response and carbon source switching

Examination of the genes that were fully overexpressed or underexpressed revealed several themes. First, the differentially expressed genes were enriched for sRNAs and genes of the PE/PPE family, including nine overexpressed PE_PGRS genes (S9 Fig). Second, six of the under-expressed genes (*ppe50*, *ppe51*, *fadD26*, *ppsA*, *Rv0885*, and *Rv3137*) were reported to be negatively regulated by the stress-responsive alternative sigma factor SigM, while one of the overexpressed genes (*Rv3093c*) was reported to be positively regulated by SigM [59]. This suggests that there may be increased SigM activity in $\Delta rnj_{H37Rv}$, although the *sigM* gene itself was not increased at the transcript level. Finally, the differentially expressed genes included several associated with hypoxia responses (the sRNAs MTS2823 and F6, and the protein-coding genes *fdxA*, *ppe31*, *clgR*, and *Rv3740c*) and several associated with utilization of various carbon sources (*ppe50*, *ppe51*, *mcm1C*, *prpC*, *Rv1066*) [60–68], suggesting that the metabolic status of the $\Delta rnj_{H37Rv}$ may differ from that of $WT_{H37Rv}$.

### Overexpression of the sRNAs *Mts2823* and *Mcr11* is necessary but not sufficient for INH tolerance in Δ*rnj* Mtb

The sRNAs *Mts2823* and *Mcr11* were two of the most overexpressed genes in the $\Delta rnj_{H37Rv}$ strain (S4 Table). Since sRNAs have been implicated in adaptation to different stresses, we sought to investigate if their increased expression in Δ*rnj* contributed to drug tolerance. We therefore deleted each of the two sRNAs in both the $WT_{6230}$ and $\Delta rnj_{6230}$ strains and performed time-killing curves in presence of RIF or INH. We found that deletion of either of these sRNAs in the $\Delta rnj_{6230}$ background decreased INH tolerance to levels near the $WT_{6230}$ (Fig 5A, 5B, 5D and 5E). In contrast, deletion of these sRNAs had no effect on INH tolerance in the $WT_{6230}$ background (Fig 5C and 5F). Thus, *Mts2823* and *Mcr11* are necessary for the INH tolerance conferred by loss of RNase J in the $\Delta rnj_{6230}$ strain. To determine whether either sRNA was sufficient for INH tolerance, we then constructed strains overexpressing either $Mts2823$ ($Mts2823_{OE}$) or $Mcr11$ ($Mcr11_{OE}$) in the $WT_{6230}$ background. However, we found that both showed drug sensitivity levels comparable to that of $WT_{6230}$ (Fig 5C and 5F), suggesting that overexpression of either *Mts2823* or *Mcr11* alone is not sufficient to increase drug tolerance. No consistent effects were observed for strains with deletions or overexpression of these sRNAs in RIF, suggesting that effectors acting downstream of RNase J deletion may act in part via drug specific mechanisms (S10 Fig).

### Downregulation of *ppe50-ppe51* is necessary for the INH and RIF tolerance phenotypes of Δ*rnj* Mtb, and deletion of *ppe50-ppe51* confers drug tolerance to a WT strain

Two of the most strongly downregulated genes in the $\Delta rnj_{H37RV}$ strain were *ppe50* and *ppe51*, which are expressed in an operon and have previously been implicated in drug sensitivity [51,52,69]. To test the hypothesis that downregulation of *ppe50* and *ppe51* contributes to the drug tolerance of the Δ*rnj* strains, we ectopically overexpressed them in the $\Delta rnj_{6230}$ background and performed drug killing experiments. Overexpression of the *ppe50-ppe51* operon in the $\Delta rnj_{6230}$ strain impaired survival of Mtb in the face of both INH and RIF, producing

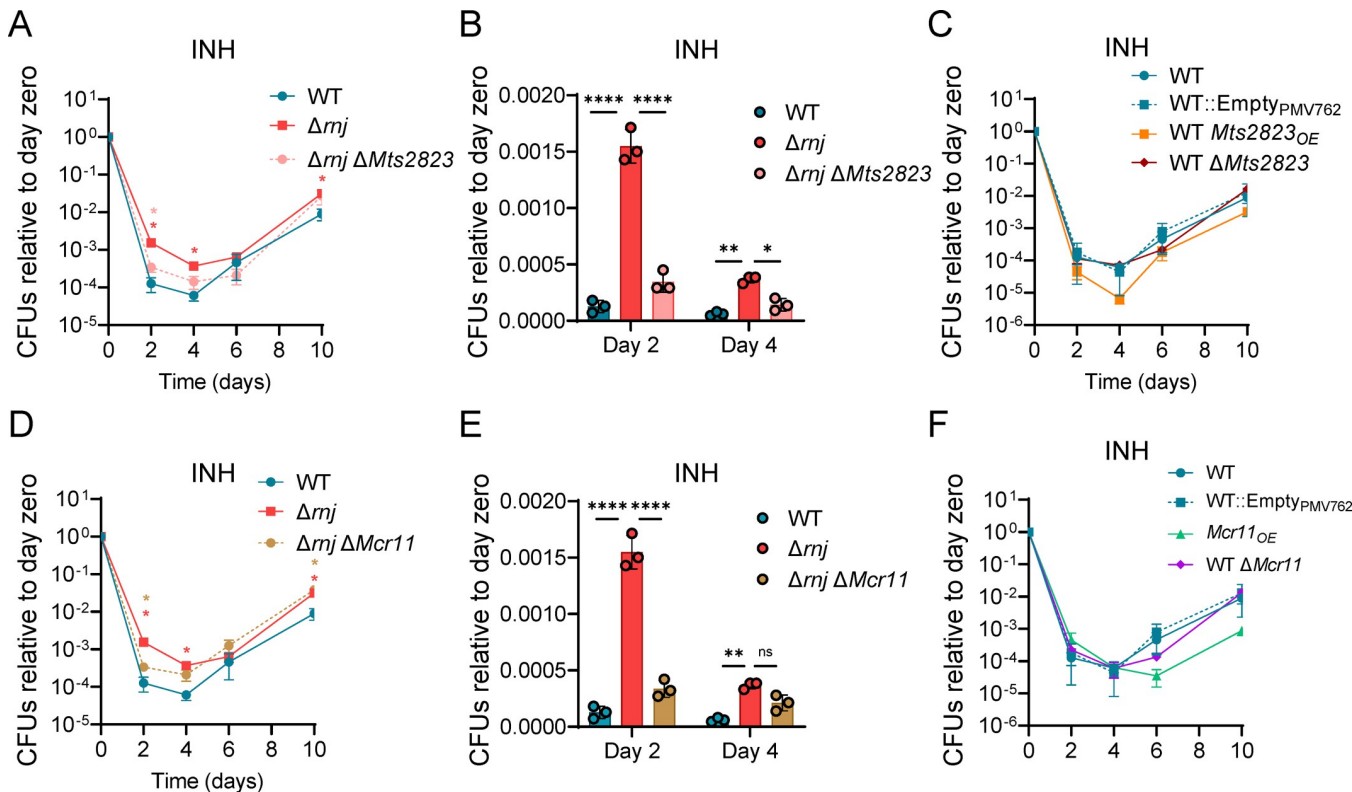

**Fig 5. Overexpression of the sRNAs *Mts2823* and *Mcr11* is necessary but not sufficient for INH tolerance in Δ*rnj* Mtb.** Time-killing curves in presence of INH (2.4 μg/mL) for strains with deletion or overexpression of *Mts2823* (**A-C**) or *Mcr11* (**D-F**) strains are shown. *$p<0.05$, **$p<0.001$ two-way ANOVA. Pink stars: comparison of Δ*rnj* Δ*Mts2823* to Δ*rnj*. Red stars: comparison of WT to Δ*rnj*. Tan stars: comparison of Δ*rnj* Δ*Mcr11* to Δ*rnj*.

drug sensitivity levels similar to those observed in the WT strain (Fig 6A). Thus, the dysregulated expression of *ppe50-ppe51* is also necessary for the altered drug susceptibility associated with loss of *rnj*. To test if loss of *ppe50-ppe51* is sufficient to confer drug tolerance, we deleted the *ppe50-ppe51* operon from $WT_{6230}$ and measured killing by INH and RIF (Fig 6B). Δ*ppe50-ppe51*$_{6230}$ displayed increased tolerance to both drugs compared to $WT_{6230}$, despite *rnj* being intact. Normal drug sensitivity was restored when we complemented Δ*ppe50-ppe51*$_{6230}$ with ectopically expressed *ppe50-ppe51*. Loss of *ppe50-ppe51* in a WT background therefore appears to be sufficient to confer tolerance to both RIF and INH.

## Discussion

Previous studies have shown that mutations in the RNase J-encoding gene were more prevalent in drug resistant Mtb clinical isolates compared to drug-sensitive isolates [16–18]. Here we show that loss of RNase J increases drug tolerance, an advantageous trait to overcome drug stress during TB treatment that could promote further development of high-level drug resistance in clinical settings. We have furthermore defined a role for RNase J in mycobacterial mRNA metabolism as a specialized degradation factor.

Despite the non-essentiality of RNase J in mycobacteria, a recent study showed that RNase J is a major component of the degradosome in Mtb [28]. Previous work also implicated RNase J as having roles in rRNA processing in *M. smegmatis* [23]. In *M. abscessus*, deletion of the *rnj* gene (MAB_3083c) resulted in changes in colony morphology, biofilm formation, and sliding motility [70]. However, the role of this nuclease in mycobacterial mRNA metabolism

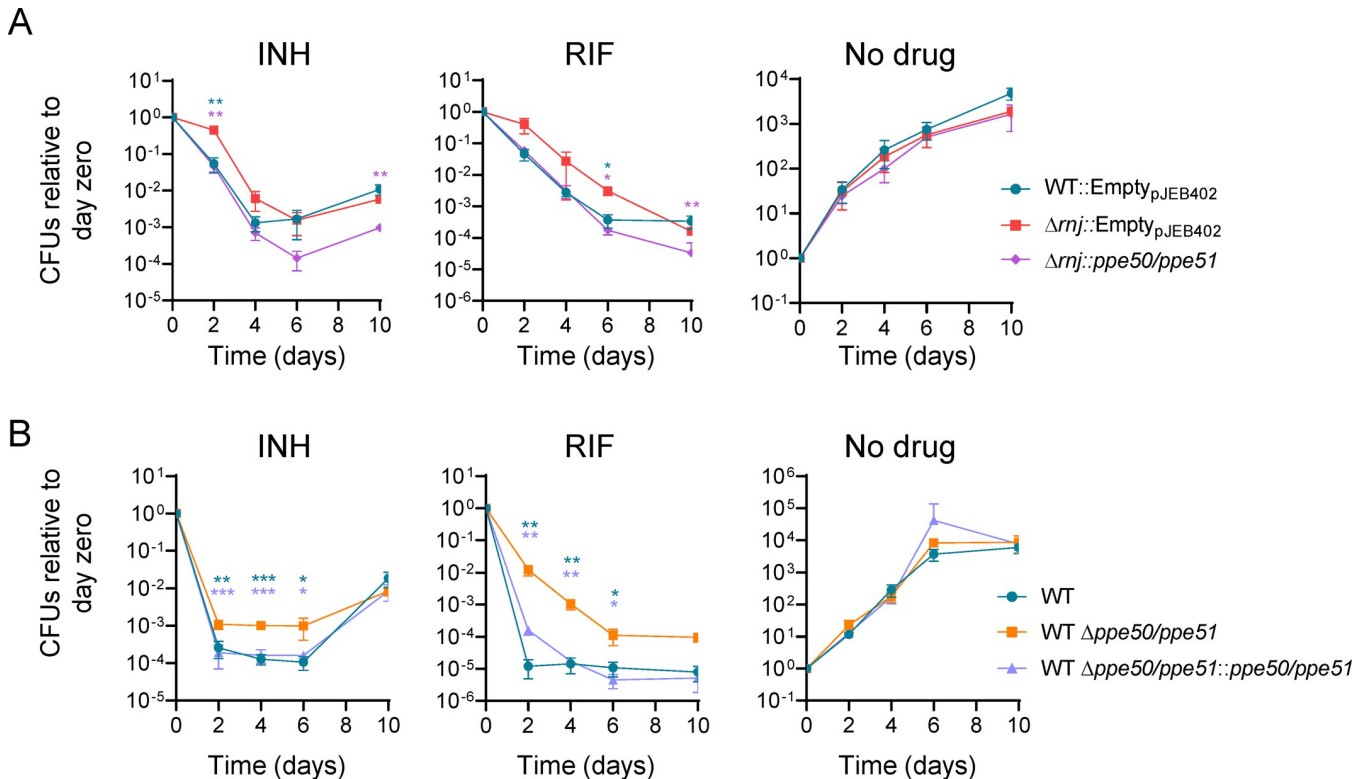

**Fig 6. Downregulation of *ppe50-ppe51* is required for the drug tolerance phenotype of Δ*rnj* Mtb, and deletion of *ppe50-ppe51* is sufficient to induce drug tolerance in a WT background.** Time-kill curves in the presence of RIF (0.6 μg/mL) or INH (2.4 μg/mL) in Mtb mc²6230 strains. **A.** *ppe50-ppe51* was ectopically expressed from a strong constitutive promoter in the Δ*rnj* strain. $^*p<0.05$, $^{**}p<0.01$, two-way ANOVA. Blue stars: comparison of WT::Empty$_{pJEB402}$ to Δ*rnj*::Empty$_{pJEB402}$. Magenta stars: comparison of Δ*rnj*::*ppe50/ppe51* to Δ*rnj*::Empty$_{pJEB402}$. **B.** *ppe50-ppe51* was deleted from the WT strain and then ectopically expressed from a strong constitutive promoter. $^*p<0.05$, $^{**}p<0.01$ $^{***}p<0.001$, two-way ANOVA. Blue stars: comparison of WT to WT Δ*ppe50-ppe51*. Lavender stars: comparison of WT Δ*ppe50/ppe51*::*ppe50/ppe51* to WT Δ*ppe50-ppe51*. FDR 0.05 (Benjamini and Hochberg) for all comparisons.

remained elusive. Our finding that deletion of RNase J affects expression of a limited number of genes is consistent with the near-wildtype growth characteristics of the mutant. Taken together with the essentiality of RNase E in mycobacteria, our results point to RNase E as playing a more important role in bulk mRNA decay. This is consistent with work in two other species that encode both RNase E and RNase J, *Rhodobacter sphaeroides* and *Synechocystis* sp. PCC6803, where the respective roles of the two nucleases have been investigated [26,71]. In contrast, RNase J plays a leading role in bulk mRNA degradation in several species that naturally lack RNase E, such as *B. subtills*, *C. diphtheriae*, *S. aureus* and *H. pylori* [72–75].

The previously reported role of RNase J in maturation of the 5' end of the 23S rRNA in *M. smegmatis* [23] was confirmed here in Mtb. In the absence of RNase J, the great majority of 23S rRNA had an additional 15 nt on its 5' end compared to the WT and complemented strains. The impact of this 5' extension is unknown. The 5' end of the 23S rRNA extends from the external face of the 50S ribosome roughly opposite from the side where the 30S subunit binds [76]. It is adjacent to ribosomal protein L13, which is required for the early stages of 30S ribosome assembly [77]. Future work should therefore investigate the impact of the 15 nt 5' extension on Mtb ribosome assembly.

Our analyses allowed us to identify native RNase J targets in Mtb as having higher G+C content and stronger predicted secondary structure than non-targets. It is important to note that the mRNA fragments we identified as direct RNase J targets generally had longer half-

lives than other fragments of the same transcript even when RNase J was present (Fig 4D, comparison of region 1 to region 2 in WT strain), in agreement with the idea that RNase J targets RNAs that are refractory to degradation by the bulk degradation machinery. This role for mycobacterial RNase J is consistent with a recently described duplex-unwinding activity in the archaeal mpy-RNase J, which was shown to degrade highly structured RNAs *in vitro* [78].

We demonstrated here that deletion of *rnj* reduces bacterial killing when Mtb is exposed to lethal concentrations of several drugs and showed that these observations are likely due to an increase in drug tolerance rather to a higher formation of persister cells. The drug tolerance is explained at least in part by gene expression changes. The overexpression of two PE/PPE genes that were downregulated in Δ*rnj*, *ppe50-ppe51*, restored the RIF and INH sensitive phenotype in the RNase J mutant to the WT levels, demonstrating that downregulation of these genes is necessary for drug tolerance in Δ*rnj*. Deletion of *ppe50-ppe51* in a WT background conferred levels of INH and RIF tolerance similar to that seen in the Δ*rnj* strain, indicating that loss of these genes is also sufficient to increase drug tolerance. With respect to INH, this finding is consistent with previous studies reporting that deletion of *ppe51* lead to increased bacterial survival in INH-treated mice [51,52] and reduced sensitivity to INH in vitro [52,69]. The relationship between *ppe51* and RIF sensitivity appears to be more complex. Deletion of *ppe51* was previously found to cause a small but significant increase in the MIC for RIF *in vitro*, but led to increased RIF sensitivity in mice [52]. The impact of *ppe51* on RIF sensitivity may therefore be condition-dependent.

It is possible that the effects of *ppe50-ppe51* on drug sensitivity are related to carbon metabolism. Dechow and collaborators reported that specific PPE51 variants promoted glycerol uptake and prevented growth arrest in acidic conditions when glycerol was the sole carbon source [62]. On the other hand, knockdown of *ppe51* affected uptake of disaccharides and attenuated Mtb growth in minimal media with disaccharides as the sole carbon source [65]. There is an increasing body of literature implicating glycerol metabolism as a pathway that affects sensitivity of Mtb to various drugs [6,51,69]. It is therefore conceivable that reduced expression of *ppe51* in the Δ*rnj* strains leads to INH tolerance through a mechanism related to glycerol metabolism.

Deletion of two non-coding sRNAs overexpressed in the mutant, *Mts2823* and *Mcr11*, in the Δ*rnj* background also showed partial restoration of drug sensitivity, suggesting that RNase J likely modulates drug tolerance via multiple mechanisms. *Mts2823* is an sRNA orthologous to the *E. coli* 6S RNA, which interacts with the RNA polymerase core preventing gene expression in mycobacteria and is highly expressed in stationary phase [79,80]. Overexpression of *Mts2823* was shown to have a slight effect on the growth rate in Mtb [79], consistent with our observations for the $Mts2823_{OE}$ strain in absence of drug. *Mcr11* is highly expressed during mouse infection [81]. Interestingly, the expression of this sRNA was increased for only ∼80% of the transcript sequence, with the 3' end region showing similar levels as in the WT strain (S11 Fig), indicating that RNase J could be involved in maturation of the 3' end of the transcript. However, more work needs to be done to understand the mechanisms of such regulation.

Taken together, our results suggest a scenario in which RNase J activity affects expression of multiple genes that together affect drug tolerance. Some of these expression changes have relatively well-delineated impacts (*e.g.*, loss or reduction of *ppe51* expression consistently causes reduced INH sensitivity in our work and that of others), while others appear to have different effects in WT and Δ*rnj* backgrounds. These findings, together with our observation that Δ*rnj* strains accumulate mRNA degradation intermediates, are consistent with the idea that mutations in *rnj* have pleiotropic effects on the physiology of Mtb and may therefore be selected *in vivo* in response to a variety of pressures.

## Supporting information

**S1 Table. Strains used in this study.**
(XLSX)

**S2 Table. Primers used in this study.**
(XLSX)

**S3 Table. Mutations in Rv2752c found in Mtb clinical isolates reported in Hicks et al., 2018 [16].**
(XLSX)

**S4 Table. DEseq2 analysis of RNAseq expression libraries.**
(XLSX)

**S5 Table. Approximate coordinates of overexpressed regions in genes labeled as partially upregulated in S4 Table.**
(XLSX)

**S6 Table. Coverage data from 5'-end-directed RNAseq.**
(XLSX)

**S1 Fig. Bioinformatic analysis of RNASeq data.**
(PDF)

**S2 Fig. Multiple sequence alignment of RNase J from *M. tuberculosis* (Mtb), *M. smegmatis* (Msm), *Streptomyces coelicolor* (Sco), *Thermus thermophilus* (Tth), and *Deinococcus radiodurans* (Dra).**
(PDF)

**S3 Fig. Loss of RNase J affects drug sensitivity in Mtb H37Rv.**
(PDF)

**S4 Fig. Growth kinetics in drug-free 7H9 and minimal media and time-kill curves in minimal media.**
(PDF)

**S5 Fig. RNase J contributes to 23S rRNA maturation in Mtb.**
(PDF)

**S6 Fig. Genes affected by loss of *rnj* are inversely affected by *rnj* overexpression.**
(PDF)

**S7 Fig. Some of the genes fully overexpressed in Δ*rnj* do not display increased stability.**
(PDF)

**S8 Fig. Decay curves used to calculate half-lives in S7 Fig.**
(PDF)

**S9 Fig. Genes that are differentially expressed in the absence of *rnj* in the H37Rv background are enriched for stable RNAs and PE/PPE family genes.**
(PDF)

**S10 Fig. The sRNAs *Mts2823* and *Mcr11* do not affect RIF tolerance in WT or Δ*rnj* Mtb.**
(PDF)

**S11 Fig. RNAseq coverage plots of sRNAs *Mts2823* and *Mcr11*.**
(PDF)

## Acknowledgments

We thank members of the Fortune and Shell labs and members of the Pathway Analysis in Tuberculosis P01 group for helpful feedback and discussions.

## Author Contributions

**Conceptualization:** Maria Carla Martini, Nathan D. Hicks, Sarah M. Fortune, Scarlet S. Shell.

**Formal analysis:** Maria Carla Martini, Nathan D. Hicks, Junpei Xiao, Scarlet S. Shell.

**Funding acquisition:** Sarah M. Fortune, Scarlet S. Shell.

**Investigation:** Maria Carla Martini, Nathan D. Hicks, Maria Natalia Alonso, Thibault Barbier, Jaimie Sixsmith.

**Methodology:** Maria Carla Martini, Nathan D. Hicks, Junpei Xiao, Scarlet S. Shell.

**Project administration:** Sarah M. Fortune, Scarlet S. Shell.

**Supervision:** Sarah M. Fortune, Scarlet S. Shell.

**Writing – original draft:** Maria Carla Martini.

**Writing – review & editing:** Maria Carla Martini, Nathan D. Hicks, Sarah M. Fortune, Scarlet S. Shell.

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
