## [Decision Letter · Decision Letter 0]

17 Mar 2022

Dear Shell,

Thank you very much for submitting your manuscript "Loss of RNase J leads to multi-drug tolerance and accumulation of highly structured mRNA fragments in Mycobacterium tuberculosis" for consideration at PLOS Pathogens. As with all papers reviewed by the journal, your manuscript was reviewed by members of the editorial board and by several independent reviewers. In light of the reviews (below this email), we would like to invite the resubmission of a significantly-revised version that takes into account the reviewers' comments.

The broad impact of the findings are somewhat limited by a lack of clear mechanism. Nevertheless, the global changes affected by RnaseJ are important and the manuscript could be strengthened by a more rigorous analysis of the RNA segments and the query about the associated half-lives. The ppe50-ppe51 needs the relevant control in the wild-type background. The authors also need to analyze the mutations observed in the clinical strains (predicted effect on the enzyme, association with known phenotype of the strain)

We cannot make any decision about publication until we have seen the revised manuscript and your response to the reviewers' comments. Your revised manuscript is also likely to be sent to reviewers for further evaluation.

Sincerely,

Helena Ingrid Boshoff

Associate Editor

PLOS Pathogens

JoAnne Flynn

Section Editor

PLOS Pathogens

Kasturi Haldar

Editor-in-Chief

PLOS Pathogens

orcid.org/0000-0001-5065-158X

Michael Malim

Editor-in-Chief

PLOS Pathogens

orcid.org/0000-0002-7699-2064

The broad impact of the findings are somewhat limited my a lack of clear mechanism. Nevertheless, the global changes affected by RnaseJ are important and the manuscript could be strengthened by a more rigorous analysis of the RNA segments and the query about the associated half-lives. The ppe50-ppe51 needs the relevant control in the wild-type background. The authors also need to analyze the mutations observed in the clinical strains (predicted effect on the enzyme, association with known phenotype of the strain)

Reviewer's Responses to Questions

**Part I - Summary**

Reviewer #1: Although there is a growing body of literature on the mechanisms of antibiotic tolerance in model organisms, the specific mechanisms mediating drug tolerance in M. tuberculosis are still poorly understood. In this work, Martini, Hicks and colleagues investigate the role of RNAseJ in multi-drug tolerance in Mtb. Bacterial GWAS studies from the authors and other groups had previously identified (probable loss of function) mutations in RNAseJ as being associated with drug-resistance. Furthermore, Tnseq analyses by Sassetti and colleagues had shown that transposon insertions in RNAseJ are associated with drug tolerance in vitro and in animal infection. Here, the authors verify that lack of RNAseJ, or replacement with a catalytically null mutant is associated with tolerance of Mtb to multiple antibiotics in vitro. They confirm that RNAseJ processes 23S rRNA. Deletion results in up/downregulation of a fairly discrete set of genes including some PPE genes and sRNAs. The authors suggest RNAseJ ‘client mRNAs’ have high GC content and more stable secondary structures. Surprisingly, measuring the half-life of several candidate RNAseJ mRNAs doesn’t show alteration of half-life +/- RNAseJ. Mechanistically, the most interesting experiments are that deletion of two sRNAs, Mts2823 and Mcr11 rescued the RNAse J tolerance phenotype to INH but not RIF, but had no phenotype in WT bacilli. Similarly, since expression of ppe50-ppe51 is decreased in the KO, the showed that OE of the two genes increased drug susceptibility of the deletion mutant to wild-type levels.

Overall, this a carefully performed study that advances mechanistic study of drug tolerance in Mtb and makes a substantial advance for the field. However, a few points of interpretation may require further nuance, and/or lack a critical control.

Reviewer #2: This study verifies the predicted role of Mtb RNaseJ in 23S rRNA maturation and degradation of more stable RNA structures; and provides new evidence for its role in drug susceptibility. Although specific mechanisms are still unknown (e.g., how is SigM activated in the rnj deletion mutant, why are sRNAs specific for INH, how exactly the loss of RNase J increases drug tolerance), this well-presented and rigorously conducted work provides good foundation for future studies.

Reviewer #3: This manuscript investigates the role of RnaseJ in Mtb in drug tolerance by characterizing an RnaseJ deletion strain of H37Rv and an auxotrophic strain of Mtb. The motivation for the study is the prior finding that WGS of clinical Mtb strains has revealed mutations in the gene encoding RnaseJ in drug resistant strains. The RnaseJ ko has the following phenotypes:

1) Loss of RnaseJ attenuates the rate of killing of Mtb by INH, Rif, Oflox, and ethambutol and these phenotypes are complemented by the wild type gene. The magnitude of the effect is most dramatic with INH at early time points.

2) Small colony size of the RnaseJ KO

3) RNA sequencing of RnaseJ ko revealed transcripts that are altered, and mapping of the RNAseq reads reveals that specific regions of transcripts are overabundant, rather than entire mRNAs, and these regions have different predicted secondary structure.

4) The authors go on to interrogate several differentially abundant transcripts and sRNAs and show suppression of the antibiotic sensitivity phenotype for some, but overexpression of these targets was not sufficient to produce drug sensitivity.

Major Comments:

Experimental

Overall, the study is rigorously performed, and I have very few experimental criticisms except for some minor technical points given below. Appropriate controls and statistical tests are applied, and the scientific quality is very high. The major experimental challenge is the magnitude of some of the phenotypes observed, which in many cases are subtle. The most significant antibiotic sensitization is to INH, but in all cases the effects are mild but statistically significant.

The effects of RnaseJ loss of RNA stability are convincing, but in most cases shown are due to persistence of a certain segment of the RNA within the transcript. Although this is plausibly attributed to the structure of that region, I think this analysis could be more extensive. The overabundant regions are quite short in most cases. Are there secondary structures of the RNAs that can be predicted in those segments? Have unannotated small RNAs been excluded as the cause of these overabundant regions as several other annotated sRNAs are found?

Given that most of the RnaseJ mutations found in clinical strains are not predicted to be null, it would be useful to engage in some structural modeling of the mutations to attempt to predict their effect.

Interpretation of findings

The major challenge of distilling the findings into a model of how RnaseJ affects antimicrobial sensitivity is the pleiotropic nature of the effects of RnaseJ deletion. This is not really a criticism, but a reflection of attributing specific mechanisms to pathways that are global. As such, the findings do not yield a clear picture of what downstream pathways are controlling this antibiotic phenotype. It is possibly a reflection of a globally altered cell, including possible changes in permeability due to cell wall alteration, metabolic changes, etc. I am sympathetic to the challenge, but the firmest conclusion is that loss of RnaseJ globally affects antibiotic sensitivity, without a clear mechanism of how this occurs. This lack of downstream mechanism somewhat limits the broad impact of the findings.

**Part II – Major Issues: Key Experiments Required for Acceptance**

Reviewer #1: 1. I’m not sure I agree with the interpretation with regards to whether the deletion mutant phenotype isn’t via lag phase modulation or doesn’t influence persister levels. As per the classic Balaban paper (PMID: 25043002), which incidentally, hasn’t been cited, lag time is strongly associated with persistence. Furthermore, the growth curve in Fig. 2A, does seem to show a difference in growth rates (although subtle, it is a log10 scale). The experiments meant to exclude lag phase aren’t totally convincing: The expt in 2D goes some way to address the issue, but even at day 4, the OD was still only ~0.01 and not really log phase. Ideally, repeating a kill curve when cultures are > OD 0.1 would help resolve the issue.

2. A puzzling result for me is with regards to the upregulation of certain transcripts in the deletion mutant. The authors nicely show that RNAseJ targets high GC-rich transcripts, which nicely fits the upregulated transcripts. Rather surprisingly, however, the half-life of some of these transcripts is not altered in the deletion mutant. That doesn’t make sense to this reader. Are the data for Fig. 3E/F for all upregulated transcripts, or for just the partially upregulated (which appear to be degraded by RNAseJ)? One would expect the partially upregulated transcripts to have a different GC content to the fully upregulated transcripts (i.e. those whose half-life is not altered).

3. The ppe50-ppe51 experiment lacks a control. Much like the former experiment with the sRNAs, the drug tolerance should also be performed in OE on WT background. Otherwise the statements on lines 361/399 that the phenotype in part explains drug tolerance in absence of RNAseJ cannot be made. Indeed, the Bellerose data suggest that this phenotype doesn’t require RNAseJ deletion.

Reviewer #2: None

Reviewer #3: I think the aspects of the paper that could be improved are the analysis of the possible structures of the RNA segments and more information about the predicted effect of RNAseJ mutations in clinical strains, but these experiments are secondary to the editorial judgement about whether the findings are of broad interest given the limitations noted above.

**Part III – Minor Issues: Editorial and Data Presentation Modifications**

Reviewer #1: 1. The lack of full processing of 23S rRNA was intriguing. I’m wondering whether at least some of the phenotype is via effects on translational speed. Addressing this experimentally is beyond the scope of the current work, but may be worth discussing in the Discussion?

2. In the text, Fig. 4 is called out as Fig. 6

3. The reference formatting is off, including, but not limited to unusually abbreviated journal names. This should be corrected.

Reviewer #2: In the RNA half-life measurements, 160 min time point was excluded because “it did not follow the initial exponential decay trend”. How many times was this experiment repeated and was this observation consistent? Is there any mechanistic explanation for this observation? It would be good to include graphs in SI that show these data.

2021 WHO TB report should be cited, not the one from 2020, since the report from 2020 shows data from year 2019, not 2020, as the authors claim in the intro. Also, TB is not top 10 cause of death anymore. The report from 2021 states that it was 13th leading cause of death in 2019 (Fig. 7), while report from 2020 shows data for year 2016, when TB was 10th (Fig. 4.14). The text should be updated to reflect newer data and more recent WHO report. Note that WHO report from 2021 was mentioned in the first paragraph, but it is not listed as a reference.

Line 63 – Period is missing before “For example…”

Line 106 – liter should be L

Line 107 – “Tween 80” instead of “tween”

Line 342 – Fig 6 should be Fig 4

Reviewer #3: Figure 1A. given the compelling finding that RnaseJ mutations are found in clinical drug resistant strains, I think this panel could be better presented. The mutation types are difficult to see.

Figure 2: the colony size phenotype: is this conserved in the non-auxotrophic strain background and is it complemented by the wild type rnaseJ?

For the complementation experiments using the H86A mutation, it should be noted that the lack of complementation, although consistent with a necessity for RnaseJ activity, could also be due to protein instability unless an antibody is available to confirm expression.

Many of the references are misformatted and missing journal names and other information.

The delta symbol in front of italicized bacterial gene names should not be also italicized

PLOS authors have the option to publish the peer review history of their article (what does this mean?). If published, this will include your full peer review and any attached files.

Reviewer #1: No

Reviewer #2: No

Reviewer #3: No
---

## [Editor Report · Decision Letter 1]

27 Jun 2022

Dear Shell,

We are pleased to inform you that your manuscript 'Loss of RNase J leads to multi-drug tolerance and accumulation of highly structured mRNA fragments in Mycobacterium tuberculosis' has been provisionally accepted for publication in PLOS Pathogens.

Best regards,

Helena Ingrid Boshoff

Associate Editor

PLOS Pathogens

JoAnne Flynn

Section Editor

PLOS Pathogens

Kasturi Haldar

Editor-in-Chief

PLOS Pathogens

orcid.org/0000-0001-5065-158X

Michael Malim

Editor-in-Chief

PLOS Pathogens

orcid.org/0000-0002-7699-2064

The authors have addressed the reviewers' concerns
---

## [Editor Report · Acceptance letter]

8 Jul 2022

Dear Shell,

We are delighted to inform you that your manuscript, "Loss of RNase J leads to multi-drug tolerance and accumulation of highly structured mRNA fragments in *Mycobacterium tuberculosis*," has been formally accepted for publication in PLOS Pathogens.

Best regards,

Kasturi Haldar

Editor-in-Chief

PLOS Pathogens

orcid.org/0000-0001-5065-158X

Michael Malim

Editor-in-Chief

PLOS Pathogens

orcid.org/0000-0002-7699-2064